# bMERB domains are bivalent Rab8 family effectors evolved by gene duplication

Amrita Rai[1], Anastasia Oprisko[1], Jeremy Campos[1], Yangxue Fu[1], Timon Friese[1], Aymelt Itzen[2], Roger S Goody[1]*, Emerich Mihai Gazdag[1]*, Matthias P Müller[1]*

[1]Department of Structural Biochemistry, Max Planck Institute of Molecular Physiology, Dortmund, Germany; [2]Center for Integrated Protein Science Munich (CIPSM), Department of Chemistry, Technische Universität München, Garching, Germany

**Abstract** In their active GTP-bound form, Rab proteins interact with proteins termed effector molecules. In this study, we have thoroughly characterized a Rab effector domain that is present in proteins of the Mical and EHBP families, both known to act in endosomal trafficking. Within our study, we show that these effectors display a preference for Rab8 family proteins (Rab8, 10, 13 and 15) and that some of the effector domains can bind two Rab proteins via separate binding sites. Structural analysis allowed us to explain the specificity towards Rab8 family members and the presence of two similar Rab binding sites that must have evolved via gene duplication. This study is the first to thoroughly characterize a Rab effector protein that contains two separate Rab binding sites within a single domain, allowing Micals and EHBPs to bind two Rabs simultaneously, thus suggesting previously unknown functions of these effector molecules in endosomal trafficking.

*For correspondence: goody@ mpi-dortmund.mpg.de (RSG); emerich-mihai.gazdag@mpi-dortmund.mpg.de (EMG); matthias.mueller@mpi-dortmund. mpg.de (MPM)

Competing interests: The authors declare that no competing interests exist.

## Introduction

Rab proteins, the biggest subfamily within the superfamily of small GTPases, are major regulators of vesicular trafficking in eukaryotic cells (*Takai et al., 2001*). Like all small GTPases, Rab proteins cycle between an inactive GDP-bound and an active GTP-bound state. The cycling is tightly regulated and mediated by two families of enzymes: guanine nucleotide exchange factors (GEFs) that catalyze the GDP/GTP exchange and GTPase activating proteins that stimulate GTP hydrolysis (*Stenmark, 2009*). Additionally, a variety of different effector proteins interact specifically with GTP-bound Rab proteins and mediate their versatile physiological roles in membrane trafficking, including budding of vesicles from a donor membrane, directed transport through the cell and finally tethering and fusion with a target membrane (*Grosshans et al., 2006*). Especially in long-distance vesicular transport processes (e.g. in neuronal axons and dendrites), directed vesicular transport along cytoskeletal tracks appears to be an obvious mechanism and, consistently, different effector proteins have been reported to link Rab proteins to the cytoskeleton (*Kevenaar and Hoogenraad, 2015*; *Horgan and McCaffrey, 2011*).

One such family of effector proteins that was reported to link Rab proteins and the cytoskeleton is the Mical (molecules interacting with CasL) family (*Figure 1*) (*Fischer et al., 2005*). Most of these Mical proteins contain an N-terminal monooxygenase domain that was reported to regulate actin dynamics via reversible oxidation of a methionine residue (*Hung et al., 2011*; *Lee et al., 2013*; *Hung et al., 2013*). Additionally, all Mical proteins except the Mical C-terminal like protein (Mical-cL) contain a calponin homology (CH) and a Lin11, Isl-1 and Mec-3 (LIM) domain that have been reported to assist the interaction with actin and other cytoskeletal proteins, respectively (*Giridharan and Caplan, 2014*). Finally, all except Mical-2 contain a C-terminal coiled-coil domain

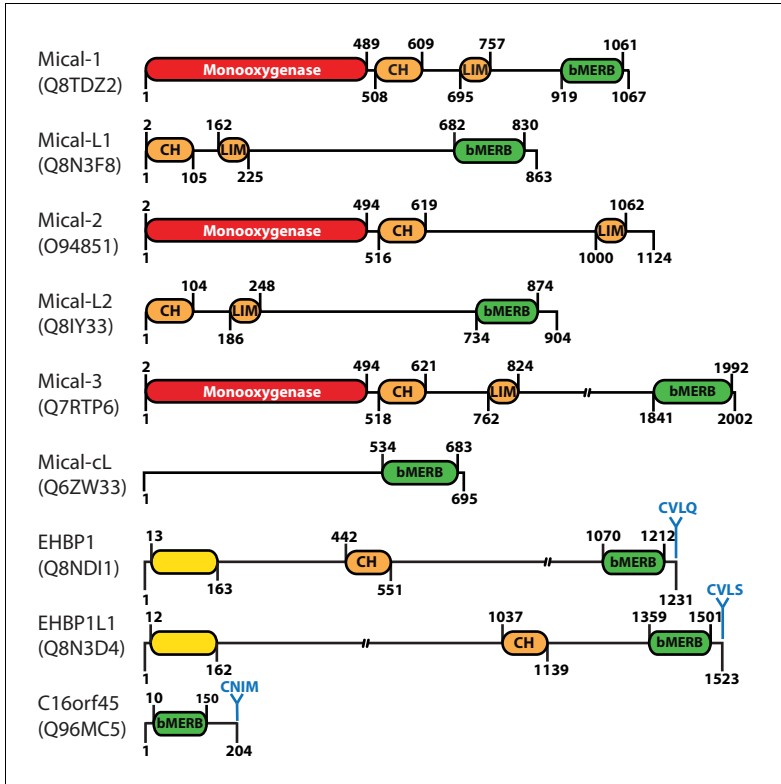

**Figure 1.** Domain architecture of human proteins containing bMERB domains. Besides their C-terminal RBD (referred to as bivalent Mical/EHBP Rab binding (bMERB) domain), most Mical proteins contain an N-terminal Monooxygenase (red), a CH- and a LIM-domain (both orange). EHBPs also contain an actin binding CH-domain and an N-terminal membrane binding C2-domain (yellow) as well as a C-terminal prenylation motif (CaaX-box) following the bMERB domain. Two proteins predicted to contain only the bMERB domains (Mical-cL and C16orf45) are also shown. For proteins with multiple known splice variants, domain boundaries are indicated for isoform 1 (Mical-1: Uniprot ID Q8TDZ2, genomic location 6q21; Mical-L1: Uniprot ID Q8N3F8, genomic location 22q13.1; Mical-L2: Uniprot ID Q8IY33, genomic location 7p22.3; Mical-3: Uniprot ID Q7RTP6, genomic location 22q11.21; Mical-cL: Uniprot ID Q6ZW33, genomic location 11p15.3; EHBP1: Uniprot ID Q8NDI1, genomic location 2p15; EHBP1L1 Uniprot ID Q8N3D4, genomic location 11q13.1). The reader is referred to the main text for further details.

that is also termed domain of unknown function (DUF) 3585 and that is known to interact with different Rab proteins (*Giridharan and Caplan, 2014*) (*Figure 1*).

According to the SMART database (*Schultz et al., 1998*), this largely uncharacterized DUF3585 domain is present in more than 450 eukaryotic proteins (including 8 human proteins, see *Figure 1*). In humans, besides the Mical proteins this includes the family of EH (Eps15-homology) domain binding proteins (EHBPs) and one uncharacterized protein (C16orf45; see *Figure 1*). Interestingly, the EHBPs also contain a CH-domain and have been described to couple vesicular transport to the actin cytoskeleton (*Shi et al., 2010*; *Guilherme et al., 2004*).

Hitherto, the structural basis and the specificity of interaction between the Rab binding domains of Micals/EHBPs and Rab proteins remained largely unknown. In this publication, we have characterized the interaction of a number of these domains with Rab proteins extensively. Our results indicate preferential binding of this family of effector proteins to Rab proteins of the Rab8 family. Additionally, the results show that at least some of these effector domains can bind two Rab proteins simultaneously, suggesting a possible role as a Rab hub in vesicular trafficking.

In order to understand the structural basis of the interaction, we solved the first x-ray crystallographic structure of the RBD from the human protein Mical-3 and the first structures of different Rab

proteins in complex with the RBDs of Mical-cL in a 1:1 stoichiometry and of Rab10 and the RBD of Mical-1 in a 2:1 stoichiometry.

This study is the first to show the structural basis of Rab proteins interacting with these RBDs and to systematically characterize the interaction with Rab proteins. Analysis of our data suggests that the second Rab binding site of these RBDs has evolved via a gene duplication event, indicating intriguing and hitherto unknown mechanisms of a concerted action of different Rab-regulated trafficking steps connected by these bivalent effector proteins, which we refer to as "bivalent Mical/EHBP Rab binding" (bMERB) domains (*Figure 1*). The study therefore substantially increases our understanding of Rab:effector interactions and will aid future research regarding the function of this diverse effector family.

## Results

### The bMERB domain preferentially binds Rab8 family proteins

Previously reported interactions of different bMERB domain containing proteins with Rab proteins included Rab1, Rab8, Rab10, Rab13, Rab15, Rab35 and Rab36 (*Giridharan and Caplan, 2014*; *Shi et al., 2010*), although not all possible combinations of the effectors and the different Rab proteins were tested nor interactions quantified. We therefore set out to systematically confirm and quantify the interaction of 5 of these Rab proteins with the bMERB domains of Mical-1, Mical-3, Mical-cL, EHBP1 and EHBP1L1 via analytical size exclusion chromatography (aSEC) (see *Table 1*, the aSEC data is shown in *Figure 2—figure supplement 1*). In these experiments, stable complex formation was detected with Rab8, Rab10, Rab13 and Rab15 (since Rab8, Rab10, Rab13 and Rab15 are closely related in amino acid sequence, we refer to them as the Rab8 family [*Klöpper et al., 2012*]). Rab1, however, failed to form stable complexes with bMERB-domain proteins, indicating low affinity.

In order to quantitatively verify the preference of the bMERB domain for Rab8 family members rather than Rab1 we performed isothermal titration calorimetry (ITC) measurements comparing the interaction of Rab1 and Rab8 with the different bMERB domains. Whereas we observed $K_D$ values of 2.2–5.2 µM for Rab1 binding to the bMERB domains of Mical proteins and no detectable binding of Rab1 to the bMERB domains of EHBPs (*Table 1* and *Figure 2a*), we detected strong binding and ca. 100 nanomolar affinities for Rab8 and the different bMERB domains (see *Table 1* and *Figure 2b*). Using Mical-cL as one representative of the bMERB family, we saw that all members of the Rab8 family bound Mical-cL with high nanomolar affinities, compared to Rab1 ($K_D$ = 5.2 µM) and Rab35 ($K_D$ = 1.8 µM; *Table 1* and *Figure 2—figure supplement 2*). The highest affinity observed was that

**Table 1.** Systematic analysis of interactions between Rab proteins and the bMERB domains of different proteins. Binding was systematically tested by analytical size exclusion chromatography (+ indicates binding in these experiments, − indicates that no complex formation was observed) and affinities were determined by ITC.

|  | Mical-1 | Mical-3 | Mical-cL | EHBP1 | EHBP1L1 |
|---|---|---|---|---|---|
| Rab1 | + | + | + | − | − |
| $K_D$ | 2.2 µM | 2.6 µM | 5.2 µM | > 10 µM | > 10 µM |
| Rab35 |  |  | + |  |  |
| $K_D$ | n.d. | n.d | 1.8 µM | n.d | n.d |
| Rab8 | + | + | + | + | + |
| $K_{D,1}$ | 55.5 nM | 27.9 nM | 253 nM | 397 nM | 159 nM |
| $K_{D,2}$ | 480 nM | 4.4 µM |  |  | 159 nM |
| Rab10 | + | + | + | + | + |
| $K_D$ | n.d. | n.d. | 790 nM | n.d. | n.d. |
| Rab13 | + | + | + | + | + |
| $K_D$ | n.d. | n.d. | 94 nM | n.d. | n.d. |
| Rab15 | + | + | + | + | + |
| $K_D$ | n.d. | n.d. | 33 nM | n.d. | n.d. |

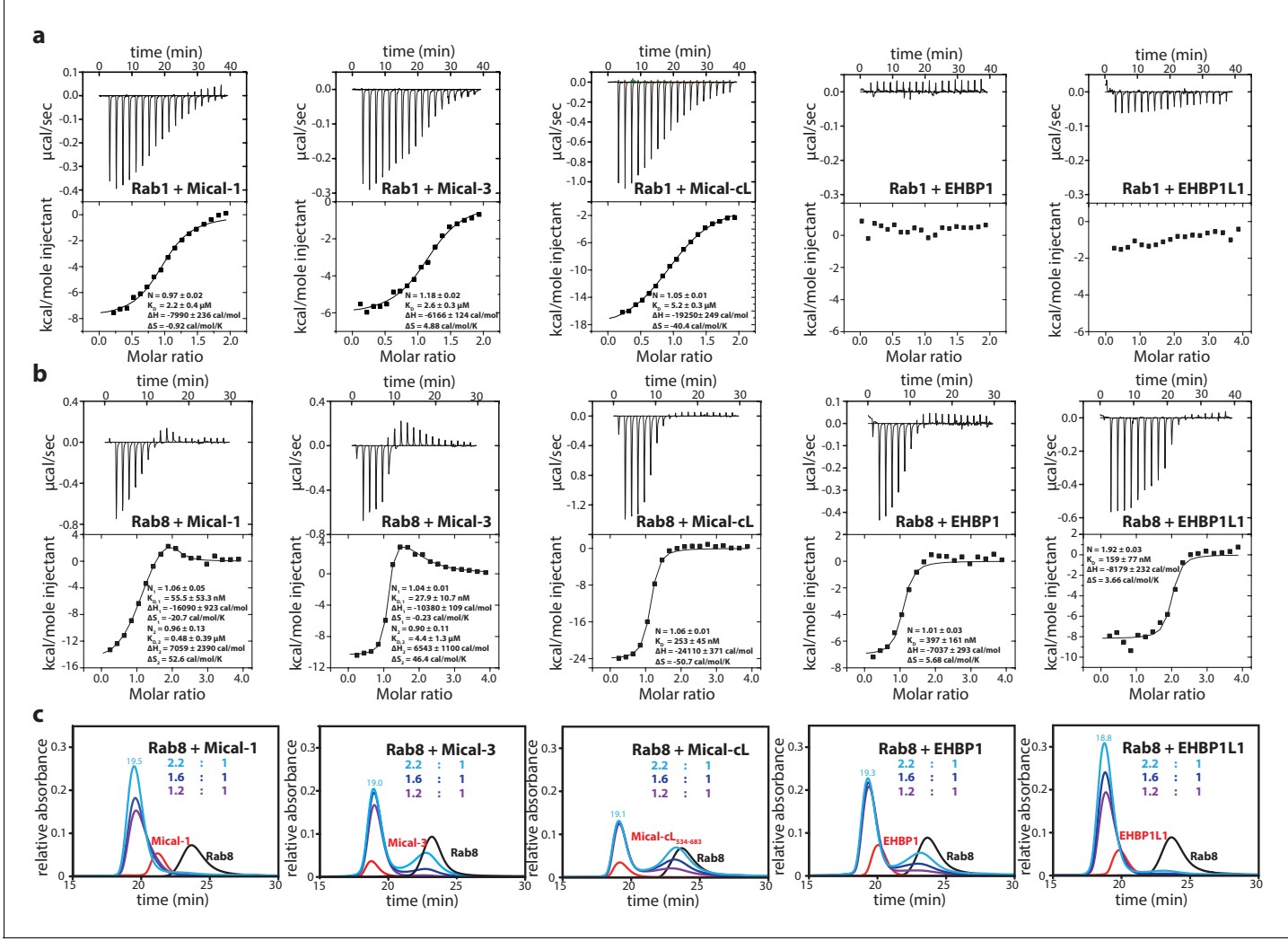

**Figure 2.** The bMERB domains preferentially interact with Rab8-family members. (a) Whereas Rab1 binds to Mical-1, Mical-3 and Mical-cL with low affinity and does not show detectable binding to EHBPs, (b) Rab8 binds with high affinity to all effector domains tested. Additionally, we observed two separate binding sites in the ITC experiments for Rab8 and Mical-1, Mical-3 and EHBP1L1 (the results of the binding fit including the stoichiometry, the $K_D$, the binding enthalpy and the binding entropy are shown within the ITC spectra). (c) Mixing different ratios of Rab8 and the RBDs (1.2:1, 1.6:1 and 2.2:1), the 2:1 stoichiometry of binding was confirmed by aSEC for Rab8:Mical-1 and Rab8:EHBP1L1, whereas a 1:1 stoichiometry was observed for Rab8:Mical-3, Rab8:Mical-cL and Rab8:EHBP1L1 as indicated by a second peak corresponding to free excess Rab8. Note that the second low affinity binding site present in Mical-3 observed via ITC could not be detected via gel filtration.

The following figure supplements are available for figure 2:

**Figure supplement 1.** Interaction of Rab proteins with the RBD of Mical-1, Mical-3, Mical-cL, EHBP1 and EHBP1L1.

**Figure supplement 2.** Interaction of Rab1, Rab35, Rab8, Rab10, Rab13 and Rab15 with Mical-cL.

**Figure supplement 3.** Formation of a ternary complex between Rab8, Rab13 and the RBD of Mical1 or EHBP1L1.

of Rab15 and Mical-cL with a $K_D$ of 33 nM. In accordance with the strong specificity of EHBPs towards the Rab8 family, Rab1a, Rab1b and Rab35 (a close relative of Rab1 which is also sometimes referred to as Rab1c) were previously shown not to interact with EHBP1/EHBP1L1 (*Shi et al., 2010*; *Nakajo et al., 2016*).

## EHBPs colocalize with Rab8-family members

Interestingly, in addition to the RBDs, EHBPs also contain CaaX-boxes at their C-termini (EHBP1: CVLQ; EHBP1L1: CVLS; *Figure 1*) for posttranslational modification with prenyl-groups. Therefore, we set out to test whether these motifs can be prenylated *in vitro* and if prenylation is responsible and necessary for correct intracellular localization. *In vitro*, both proteins can be farnesylated and geranylgeranylated by FTase and GGTase I, respectively. However, in accordance with the C-terminal amino acids of the CaaX-boxes being glutamine or serine (*Zhang and Casey, 1996*), we observed preferential farnesylation (*Figure 3a*). Using constructs containing the RBD with or without the CaaX-boxes, we next looked at the intracellular localization. These experiments clearly showed a CaaX-box dependent localization of both proteins to intracellular structures resembling endosomes (*Figure 3b*), as previously reported for the full length proteins (*Shi et al., 2010*). Additionally, both proteins showed strong colocalization with constitutively active Rab8 and Rab10 (i.e. Rab8$_{Q67L}$ and Rab10$_{Q68L}$), both known to act at endosomes (*Figure 3c*), further supporting the function of this family of effector proteins as Rab8-family binding partners.

## Some bMERB domains can bind two Rab proteins simultaneously

Besides the specificity of bMERB domains towards the Rab8 family, another interesting observation was made in the ITC experiments comparing the stoichiometry of binding of Rab8 and the different RBDs: Whereas Rab8 bound in a 1:1 stoichiometry to Mical-cL and EHBP1, a 2:1 stoichiometry binding was observed for Mical-1, Mical-3 and EHBP1L1 (*Table 1* and *Figure 2b*). For Rab8 and Mical-1/Mical-3, the ITC experiments show one high-affinity enthalpy-driven and one lower affinity entropy-driven binding site, compared to two binding sites with similar affinity for Rab8 and EHBP1L1.

In order to confirm the observed differences, we repeated the aSEC experiments with varying ratios of Rab8 and the different RBDs of Micals and EHBPs (1.2:1, 1.6:1 and 2.2:1; *Figure 2c*). These experiments clearly confirmed the aforementioned differences in the stoichiometry of binding with a 2:1 stoichiometry being observed for the Rab8:EHBP1L1 and Rab8:Mical-1 complexes, but not for others tested. The low affinity second binding site of Mical-3 (K$_D$ = 4.4 µM as determined by ITC, *Table 1*) could also not be detected in these experiments, suggesting dynamic complex formation with a large k$_{off}$. These data show that Mical-1 and EHBP1L1 contain two binding sites that bind Rab8 with high affinity, whereas Mical-3 (and possibly Mical-cL and EHBP1) contain one high affinity and a second lower affinity binding site for Rab8.

The presence of two distinct Rab binding sites on certain bMERB domains was a striking observation pointing towards a possible function of these effector proteins in sorting cargo and/or linking different endosomal trafficking pathways regulated by different Rab proteins. In accordance with this idea, recent studies on Mical-L2 dependent GLUT4 translocation showed that trafficking was dependent on a concerted action of Rab8 and Rab13 (*Sun et al., 2010*, *2016*). We consequently also tested whether the effector proteins might be able to simultaneously bind two different Rab proteins in a 1:1:1 (RabX:effector:RabY) complex using both Rab8 and Rab13 and the corresponding aSEC experiments clearly confirmed the formation of a ternary complex of Rab8:Mical-1:Rab13 as well as Rab8:EHBP1L1:Rab13 (*Figure 2—figure supplement 3*).

## The structural basis of Rab:bMERB interaction

In order to understand the mode of interaction of Rab proteins and the C-terminal Rab-binding domain of Micals/EHBPs, we first aimed at determining the structure of the RBD of one member of these families. We succeeded in crystallizing a selenomethionine derivative of Mical-3$_{1841-1990}$ containing the whole predicted bMERB domain and solved the structure with a resolution of 2.7 Å (data and refinement statistics are shown in *Table 2*).

The asymmetric unit contains two copies of Mical-3 that form a central 4-stranded coiled-coil composed of α-helices 2 and 3 of each monomer flanked by α-helices 1 on both sides (*Figure 4a*). Interactions between the monomers mainly occur via two hydrophobic patches and some additional charged interactions (*Figure 2b*). Overall, the structure shows that each monomer consists of a central helix (α2, residues K1891-R1937) and N- and C-terminal helices folding back on this central helix.

The completely α-helical fold of this protein is common to many Rab effector proteins and most of them bind the interacting Rab proteins via two α-helices (*Mott and Owen, 2015*). In order to test whether this is also true for the bMERB domains, we screened for crystallization conditions of

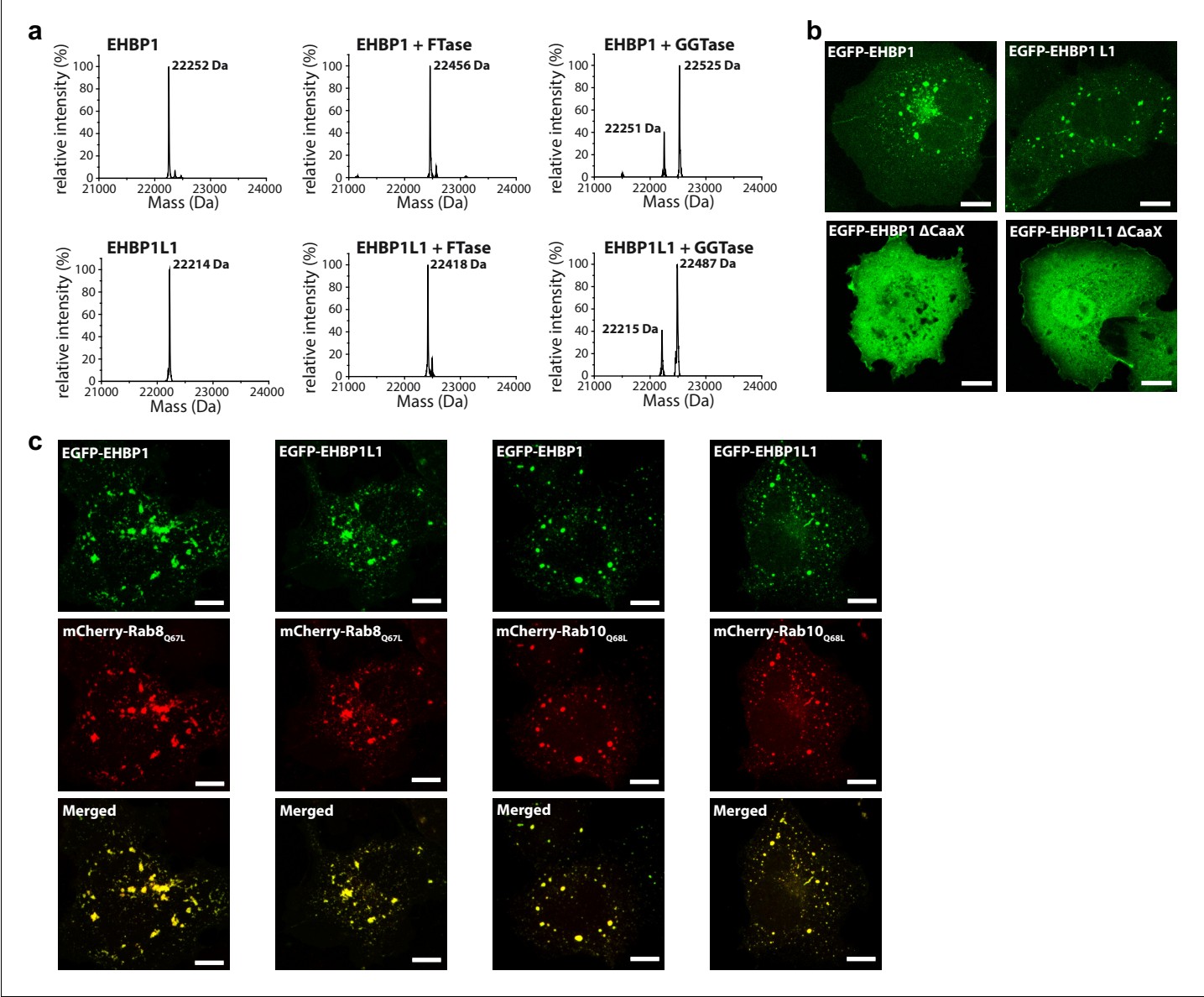

**Figure 3.** Prenylated EHBP1 and EHBP1L1 colocalize with Rab8 and Rab10. (**a**) EHBP1 and EHBP1L1 can be prenylated *in vitro* as shown by mass spectrometry. After incubation of the purified proteins including the CaaX-motifs (theoretical masses of the purified proteins: 22253.3 Da (EHBP1); 22214.0 Da (EHBP1L1); left panel) with Farnesytransferase (FTase, middle panel) or Geranylgeranyltransferase (GGTase, right panel) farnesylation/ geranylgeranylation lead to an increase in mass of 205.4/272.5 Da, respectively. Note that farnesylation, in contrast to geranylgeranylation, appears to be more efficient under similar conditions and goes to completion. This is in agreement with the sequence of the CaaX-motifs in both proteins containing a Gln/Ser at their C-terminus which has been shown to favor farnesylation. (**b**) Whereas the constructs containing the bMERB domain and the CaaX-motif (EHBP1$_{1047-1231}$, EHBP1L1$_{1340-1523}$) localize to intracellular structures resembling endosomes, deletion of the CaaX-motif (ΔCaaX) leads to a cytosolic distribution for both EHBP1 and EHBP1L1 (Scale bars: 10 μm). (**c**) Both EGFP-EHBP1$_{1047-1231}$ and EGFP-EHBP1L1$_{1340-1523}$ (upper panel) show strong colocalisation with mCherry-Rab8$_{Q67L}$ and mCherry-Rab10$_{Q68L}$ (middle panel) as indicated in the merged images (lower panel). The localization pattern resembles that of endosomes (Scale bars: 10 μm).

bMERB:Rab complexes. Well diffracting crystals were found using the RBD of Mical-cL (residues 534–683) in complex with different Rab proteins (Rab1, Rab8 and Rab10). All structures were solved using a single chain of Mical-3 and the structure of Rab1 (pdb id 3nkv) or Rab8 (pdb id 4lhw) as search models for molecular replacement (the data collection and refinement statistics are shown in *Table 2*).

**Table 2.** Data-collection and processing statistics (values in parentheses are for the outer shell).

| | SeMet Mical-3$_{1841-19902}$[†] | Rab1:Mical-cL$_{534-683}$ | Rab8:Mical-cL$_{534-683}$ |
|---|---|---|---|
| **Data collection**[*] | | | |
| X-Ray Source | X10SA at SLS | X10SA at SLS | X10SA at SLS |
| Wavelength (Å) | 0.978956 | 0.99992 | 1.00009 |
| Resolution range (Å) | 47.8–2.7 (2.8–2.7) | 45.8–2.3 (2.4–2.3) | 46.0–2.85 (2.95–2.85) |
| Space group | P 2$_1$ 2$_1$ 2$_1$ | I 2 2 2 | C 2 2 2$_1$ |
| Unit cell<br>a, b, c (Å)<br>α, β, γ (°) | 51.9, 78.8, 95.6<br>90.0, 90.0, 90.0 | 61.75, 129.38, 129.85<br>90.0, 90.0, 90.0 | 62.4, 122.4, 139.15<br>90.0, 90.0, 90.0 |
| No. of reflections<br>Total<br>Unique | 272679 (28873)<br>20544 (2119) | 308332 (36902)<br>23530 (2760) | 162311 (16579)<br>12810 (1224) |
| Multiplicity | 13.3 | 13.1 | 12.7 |
| Completeness | 99.1 (98.5) | 100.00 (100.00) | 99.9 (100.0) |
| R$_{merge}$ (%) | 13.8 (114.6) | 10.5 (74.3) | 8.9 (77.6) |
| R$_{meas}$ (%) | 14.4 (119.1) | 10.9 (77.2) | 9.2 (80.6) |
| I/σ(I) | 16.8 (3.6) | 16.27 (3.89) | 16.45 (3.19) |
| f′ / f′′ | -7.29 / 3.84 | - | - |
| **Refinement** | | | |
| Resolution range (Å) | 47.8–2.7 (2.77–2.7) | 45.8–2.3 (2.4–2.3) | 46.0–2.85 (3.07–2.85) |
| No. of reflections (work set) | 10553 | 23521 | 12808 |
| R$_{work}$ (%) | 25.1 (20.5) | 17.9 (26.9) | 23.7 (31.1) |
| R$_{free}$ (%) | 28.2 (36.2) | 20.7 (29.8) | 28.8 (35.0) |
| No. of atoms<br>Protein<br>Ligands<br>Water | 2095<br>14<br>- | 2552<br>33<br>27 | 2426<br>33<br>2 |
| B-factors<br>Protein<br>Ligands<br>Water | 72.9<br>65.2<br>- | 76.7<br>49.8<br>73.1 | 101.2<br>111.2<br>122.6 |
| R.m.s deviations<br>Bond length (Å)<br>Bond angles (°) | 0.016<br>1.809 | 0.008<br>1.104 | 0.009<br>1.175 |
| Ramachandran plot<br>Favored<br>Additionally allowed<br>Outliers | 98.8<br>1.2<br>0 | 98.4<br>1.6<br>0 | 96.1<br>3.3<br>0.7 |
| PDB entry code | 5SZG | 5SZH | 5SZI |
| | **Rab10:Mical-cL$_{534-683}$** | **Rab10$_{1-175}$:Mical-1$_{918-1067}$** | **Rab1$_{R8N}$:Mical-cL$_{534-683}$** |
| **Data collection** | | | |
| X-Ray Source | X10SA at SLS | X10SA at SLS | X10SA at SLS |
| Wavelength (Å) | 1.00009 | 0.99997 | 0.91908 |
| Resolution range (Å) | 48.2–2.66 (2.7–2.66) | 44.0–2.8 (2.9–2.8) | 44.8–2.8 (2.85–2.8) |
| Space group | P 2$_1$ 2$_1$ 2 | P 2$_1$ 2$_1$ 2$_1$ | C 2 2 2$_1$ |
| Unit cell<br>a, b, c (Å)<br>α, β, γ (°) | 153.7, 61.9, 55.6<br>90.0, 90.0, 90.0 | 58.4, 59.0, 198.2<br>90.0, 90.0, 90.0 | 62.2, 117.0, 139.4<br>90.0, 90.0, 90.0 |
| No. of reflections<br>Total<br>Unique | 187267 (6488)<br>15861 (676) | 222905 (21259)<br>17508 (1689) | 170436 (8892)<br>12904 (645) |
| Multiplicity | 11.8 | 12.7 | 19.2 |

*Table 2 continued on next page*

*Table 2 continued*

| | | | |
|---|---|---|---|
| Completeness | 99.9 (100.0) | 99.6 (99.9) | 100.0 (100.0) |
| $R_{merge}$ (%) | 13.7 (158.6) | 11.8 (72.3) | 7.6 (110.6) |
| $R_{meas}$ (%) | 14.3 (167.8) | 12.3 (75.4) | 7.9 (114.9) |
| $I/\sigma(I)$ | 12.2 (1.4) | 14.1 (3.3) | 22.8 (2.45) |
| f' / f'' | - | - | - |
| **Refinement** | | | |
| Resolution range (Å) | 48.2–2.66 (2.83–2.66) | 44.0–2.8 (2.98–2.80) | 44.8–2.8 (3.0–2.80) |
| No. of reflections (work set) | 15857 | 17499 | 12904 |
| $R_{work}$ (%) | 22.4 (30.2) | 23.7 (29.4) | 20.8 (30.8) |
| $R_{free}$ (%) | 26.6 (36.9) | 28.8 (35.6) | 26.1 (38.4) |
| No. of atoms<br>Protein<br>Ligands<br>Water | 2559<br>40<br>39 | 3676<br>66<br>4 | 2565<br>33<br>3 |
| B-factors<br>Protein<br>Ligands<br>Water | 77.8<br>77.9<br>70.2 | 88.6<br>85.4<br>68.7 | 86.2<br>81.2<br>73.8 |
| R.m.s deviations<br>Bond length (Å)<br>Bond angles (°) | 0.004<br>0.756 | 0.013<br>1.506 | 0.010<br>1.178 |
| Ramachandran plot<br>Favored<br>Additionally allowed<br>Outliers | 98.1<br>1.9<br>0 | 96.7<br>3.3<br>0 | 98.1<br>1.6<br>0.3 |
| PDB entry code | 5SZJ | 5LPN | 5SZK |

[*]All data sets were collected from one single crystal on beamline X10SA of the Swiss Light Source (Paul Scherrer Institute, Villigen, Switzerland)

[†]Data collections statistics for SAD data refer to unmerged Friedel pairs.

In all structures (Rab1:Mical-cL, Rab8:Mical-cL and Rab10:Mical-cL) we found one Rab protein bound to one molecule of Mical-cL (*Figure 5a*), in agreement with the previous observations that all Rab proteins tested bind only one site in Mical-cL. Most interactions were visible between the Rab proteins and α-helix 3 of Mical-cL with some additional contributing residues from α-helix 2, forming extensive contacts involving residues within switch I and II of the Rab proteins (*Figure 5b*). In all cases, hydrophobic interactions between the hydrophobic patch II in Mical-cL and residues from Rabs forming a hydrophobic pocket (residues Ile43, Phe70 and Ile73 in Rab8) and a triad of aromatic amino acids (Phe45, Trp62, Tyr77 in Rab8) known from all Rab:effector complexes solved to date (*Itzen and Goody, 2011*) were also observed in the Rab:Mical-cL structures.

Interestingly, the Rab-binding interface in Mical-cL has a substantial overlap with the dimer interface observed in the structure of Mical-3 above. Additionally, even though all Mical constructs used have a similar molecular weight of ~18 kDa, whereas Mical-1 runs as an apparent monomer in aSEC and binding of a Rab protein induces a clear shift to higher molecular weight, both Mical-3 and Mical-cL run as apparent dimers in aSEC and binding of a Rab protein disrupts the dimer, thus not leading to a shift in retention time upon complex formation (*Figure 2—figure supplement 1*). It is however not clear at this point whether the dimer formation of Mical-3 and Mical-cL and the disruption of the dimer upon Rab-binding is of functional significance.

The specificity of effector proteins towards certain Rab families is generally achieved via interaction with regions termed Rab subfamily motifs (RabSFs) 1–4 (*Khan and Ménétrey, 2013*; *Moore et al., 1995*; *Pereira-Leal and Seabra, 2000*). In the Rab:Mical-cL structures, we observed extensive interaction of Mical-cL with RabSF1 (Tyr6, Asp7, Leu9, Lys11 in Rab10) and (less interactions) with RabSF2 (Asp31, Ser40 in Rab10). Accordingly, the sequence alignment of different Rab proteins (*Figure 5*) shows strong conservation of the interacting amino acids within these motifs

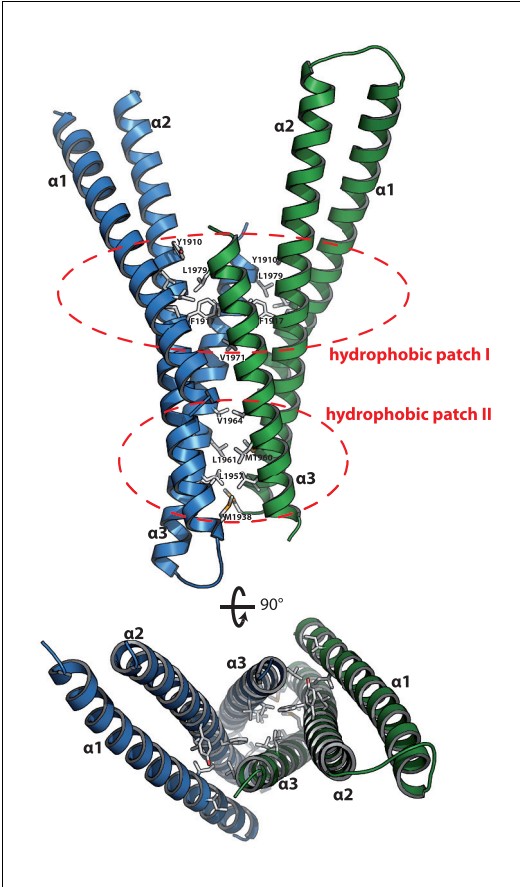

**Figure 4.** Structure of the Rab binding domain of Mical-3. Mical-3 folds into three α-helices, the central α-helix 2 and α-helices 1 and 3 folding back on the central helix. The dimer observed in the asymmetric unit is formed mostly by hydrophobic interactions involving the same hydrophobic patches in both monomers, and α-helices 2 and 3 from each monomer form a central 4-stranded coiled-coil.

amongst Rab1 (Rab1a/b, Rab35) and Rab8 (Rab8a/b, Rab10, Rab13, Rab15) family members that interact with bMERB domains, but not for other Rab proteins (a comparative scheme of the residues involved in interactions in the different complexes is shown in *Figure 5—figure supplement 1a*). Since the interacting residues in the RabSF1 and RabSF2 regions are strongly conserved between both Rab1 and Rab8 families, this did not explain the observed preference of the RBD towards the Rab8 family. However, we observed that the main chain atoms of the N-terminal residues preceding the RabSF1 motif can be traced in the electron density and seem to interact with amino acids within α-helix 1 and 2 of Mical-cL, even though the electron density in this region did not allow a precise localization of the side chains. In contrast to the main interacting helix α3, which adopts a similar position in all three structures, we observed slightly different orientations of the α-helices 1 and 2, adopting a position slightly further away from Rab1 compared to Rab8 and Rab10 (*Figure 5a*). Interestingly, whereas Rab1 contains a glutamate near the N-terminus (Glu4), all Rab8 family members contain one or (in the case of Rab10) two lysine residues in this region that point towards a negatively charged patch in Mical-cL (*Figure 5b*). Additionally, Rab35 contains an Arg residue within this N-terminal region and also displays a slightly higher affinity towards Mical-cL compared to Rab1 (*Figure 2—figure supplement 2*). We therefore tested whether these N-terminal residues of the Rab proteins might determine the specificity of bMERB domains towards Rab8 and its homologues rather than Rab1. We constructed a chimera of Rab1 containing the 4 N-terminal aa of Rab8, thus exchanging the negatively charged glutamate for a positively charged lysine. The x-ray crystallographic structure of this Rab1 chimera (termed Rab1$_{R8N}$) in complex with Mical-cL clearly showed that the helices 1 and 2 move closer and adopt a similar conformation as observed in the structures of Rab8:Mical-cL and Rab10:Mical-cL (*Figure 5a*). Additionally, ITC measurements showed that the chimera had an approximately five-fold increased binding affinity compared to Rab1 (*Figure 5c*). In contrast to Rab1, the chimera Rab1$_{R8N}$ bound both EHBP1 and EHBP1L1 in aSEC experiments (*Figure 5—figure supplement 1b*), thus clearly indicating that the N-terminus is important for the interaction and contributes to the observed specificity of bMERB domains towards Rab proteins.

## The second Rab binding site has evolved by gene duplication

Stimulated by the evidence for two Rab binding sites in some bMERB domains (Mical-1, Mical-3 and EHBP1L1), we searched for crystallization conditions of these RBDs with Rabs in a 1:2 stoichiometry. Crystallization conditions were found using a complex of Rab10$_{1–175}$ and the RBD of Mical-1 (residues 918–1067), yielding crystals that diffracted to a resolution of 2.8 Angstrom at a synchrotron X-ray source and the resulting structure indeed showed two molecules of Rab10 bound to Mical-1 (*Figure 6a*). In addition to the binding site corresponding to the one previously observed in Mical-cL, an additional binding site was identified: Whereas this site is composed of the N-terminal half of

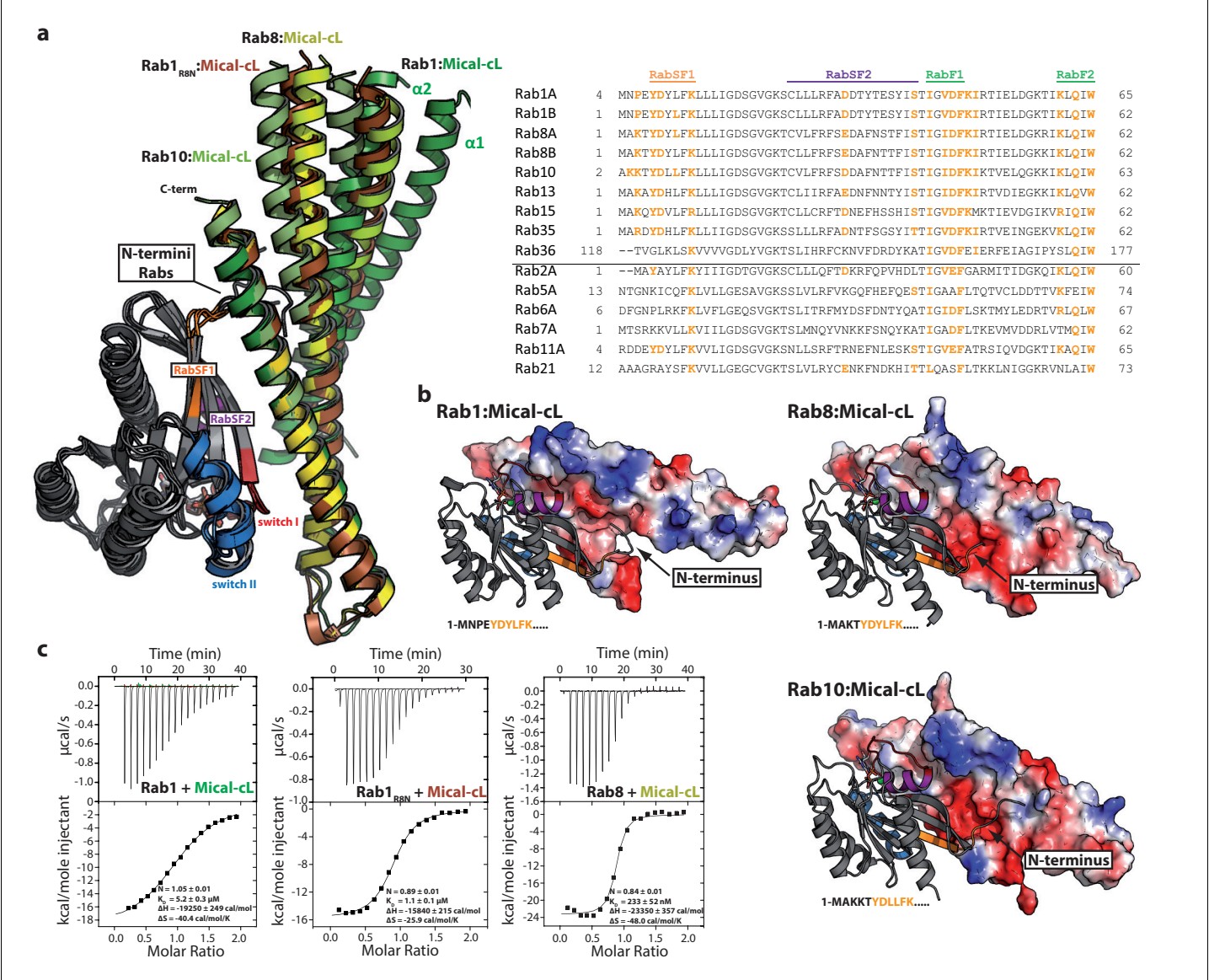

**Figure 5.** The specificity of Rab proteins binding to bMERB domains. (a) A superposition of the complex structures of Rab1:Mical-cL, Rab8:Mical-cL, Rab10:Mical-cL and Rab1$_{R8N}$:Mical-cL shows that Rabs bind Micals via their N-terminus (including RabSF1), RabSF2 as well as the switch regions (Rabs are shown in cartoon representation, switch I – red, switch II – blue, RabSF1 – orange, RabSF2, magenta; Micals are shown in cartoon representation and colored in dark green (Mical-cL interacting with Rab1), yellow (Mical-cL interacting with Rab8), light green (Mical-cL interacting with Rab10) or brown (Mical-cL interacting with Rab1$_{R8N}$). The sequence alignments of different Rab proteins clearly shows that the interacting residues of Rab proteins with Micals (red residues) are highly conserved (orange residues) in Rab1 and Rab8 family members (Rab1a, Rab1b, Rab35, Rab8, Rab10, Rab13, Rab15), but not in other Rabs (below the black line). (b) In all structures of Rab proteins in complex with Mical-cL, the N-termini of the Rab proteins point towards a negatively charged patch of Mical-cL (Rabs are shown in cartoon representation as above; the surface of Mical-cL is colored by charge, red – negative charge, blue – positive charge). The sequence of the N-terminal residues of each Rab protein is shown below the corresponding structure: Whereas Rab1 contains a negatively charged glutamate at position 4, Rab8 and Rab10 contain one or two lysine residues at position 3 or at position 3 and 4, respectively. Consistently, the negatively charged N-terminus of Rab1 seems to repel α-helices 1 and 2 of Mical-cL and they adopt a conformation slightly further away from Rab1 compared to Rab8 and Rab10 (also see (a)). However, after mutating the 4 N-terminal residues of Rab1 to the corresponding sequence of Rab8 (the resulting chimera is called Rab1$_{R8N}$), the structure of Rab1$_{R8N}$:Mical-cL shows a similar conformation of α-helices 1 and 2 as in the structure of Rab8:Mical-cL. (c) Consistently, ITC measurements show that the affinity of binding increases approximately five-fold after mutating the N-terminal residues (Rab1:Mical-cL: K$_D$ = 5.2 µM; Rab1$_{R8N}$:Mical-cL: K$_D$ = 1.1 µM; Rab8:Mical-cL: K$_D$ = 0.23 µM).

The following figure supplements are available for figure 5:

**Figure supplement 1.** The N-termini of Rabs determine the specificity towards bMERB domains.

*Figure 5 continued on next page*

*Figure 5 continued*

**Figure supplement 2.** Comparison with other Rab:effector structures.

the bMERB domain (α-helix 1 and the first half of α-helix 2), the Rab binding site observed in both Mical-1 and Mical-cL comprises the C-terminal half (second half of α-helix 2 and α-helix 3).

Upon closer inspection and alignment, a strong similarity between the two Rab binding sites in Mical-1 became obvious, involving the same/similar residues both within the two different molecules of Rab10 as well as the two binding sites in Mical-1, respectively (*Figure 6a*). An alignment of the sequences of the corresponding N- and C-terminal halves of all different Micals (Mical-1, Mical-cL, Mical-3, Mical-L1 and Mical-L2), EHBP1 and EHBP1L1 (see *Figure 6—figure supplement 1*, the example for Mical-1 is shown in *Figure 6b*) with Clustal Omega (*Sievers et al., 2011*) highlights the striking similarity between the binding sites and shows the strong conservation of Rab-interacting residues within the two binding sites. A non-exhaustive list and a close-up view of several of these interactions is shown in *Table 3* and *Figure 6—figure supplement 2*, respectively. It should be noted that use of the N- and C-terminal halves of only one of the bMERB domains was not sufficient for Clustal Omega alignment to converge and find the conserved residues within the separate halves. In contrast, the webserver HHrepID (*Biegert and Söding, 2008*) nicely predicted and aligned the two repeats present in Mical-1 with a p-value of $1.1^{-5}$.

Consistent with the localization of the two separate binding sites within the N-terminal and the C-terminal half of the bMERB domain, respectively, deletion constructs lacking either α-helix 1 (Mical$_{960-1067}$) or α-helix 3 (Mical-1$_{918-1020}$) displayed a clear 1:1 stoichiometry of Rab binding both in aSEC and ITC experiments (*Figure 6c*). Furthermore, the ITC data allowed us to clearly allocate the high affinity binding to the C-terminal binding site and the lower affinity binding to the N-terminal binding site.

In summary, the strong conservation of interacting residues between both sites as well as the structural conservation of the binding sites lead us to conclude that this family of Rab binding proteins must have evolved via gene duplication (*Figure 6d*). Furthermore the strong conservation of interacting residues not only between the two separate binding sites in Mical-1, but also between the different bMERB domains (see alignments in *Figure 6—figure supplements 1* and *3*) suggests that all of these proteins contain a second (possibly low affinity) binding site. Further analysis of the Rab specificity of both sites within these proteins will therefore be of great interest.

## Discussion

In this publication, we present a thorough biochemical and structural analysis of a Rab effector domain termed bivalent Mical/EHBP Rab binding (bMERB) domain. The results show that the domains probably constitute a Rab8-effector family involved in endosomal trafficking, and the Rab-binding specificity can be well explained from the 3-dimensional structures of complexes determined in this work. Furthermore, we show that at least some of these domains contain two separate binding sites for Rab-proteins, suggesting previously unknown functions, as discussed below. The strong similarity between the 2 binding sites within one effector domain strongly suggests an evolutionary development via gene duplication.

The high specificity of the effector domains towards Rab8 family members can be well explained from the structural analysis of Rab:bMERB complexes. Specificity-determining interactions were seen between the effector domains and the RabSF1 and RabSF2 motifs. However, additional interactions were required to increase the specificity even further, thus allowing the proteins to distinguish Rab1- and Rab8-family members. In this regard, we showed that the N-terminal residues preceding the RabSF1 motif contribute to this specificity, an observation that has previously not been made in other Rab:effector interactions. However, as alluded to in the introduction, the presence of multiple isoforms (e.g. Mical-1, Mical-L1 etc.) of the proteins, as well as the demonstrated presence of two separate binding sites, might also point towards a broader and more diverse Rab-binding spectrum and is the subject of ongoing research in our work.

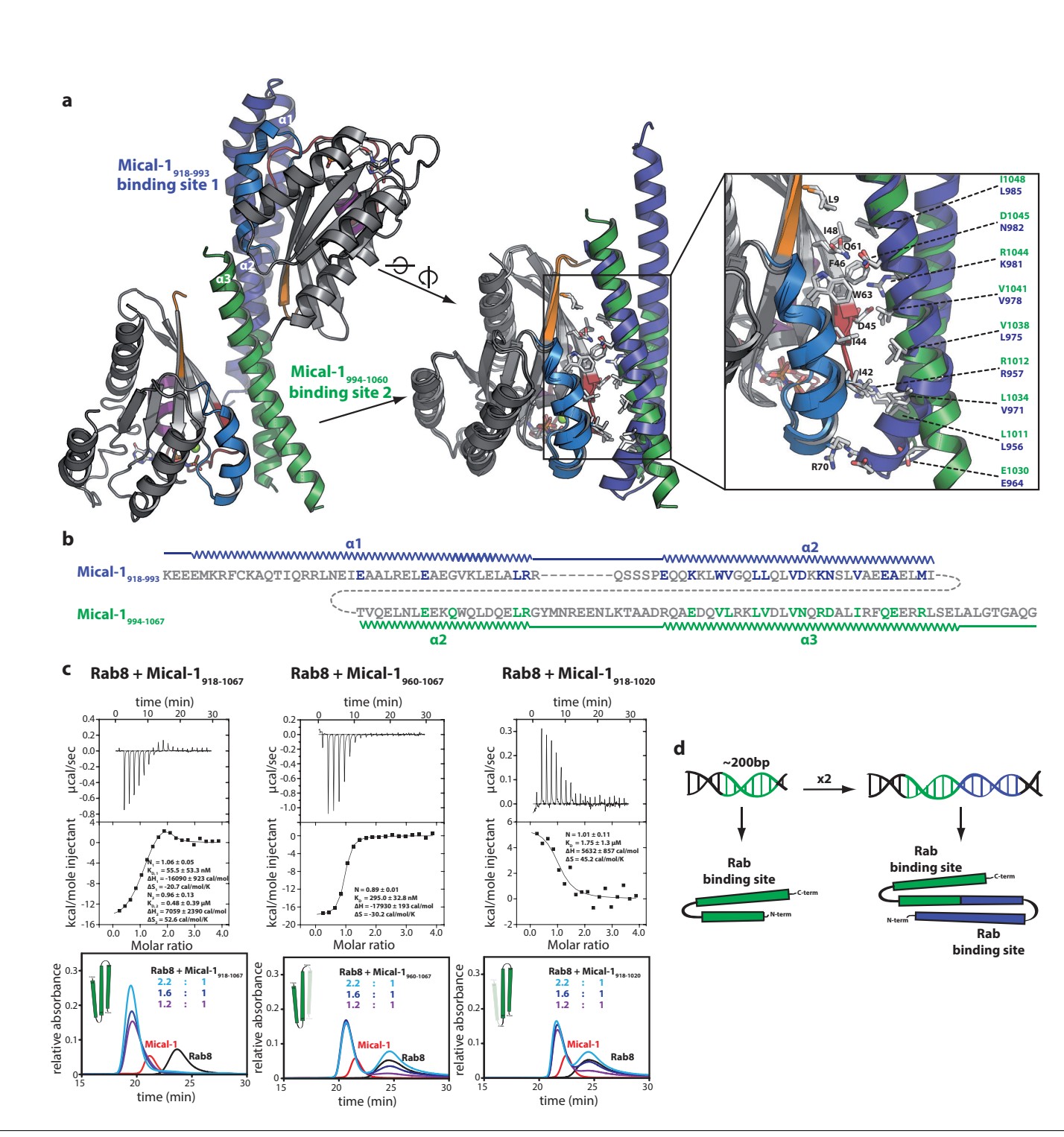

**Figure 6.** The two Rab binding sites are highly similar. (a) The structure of Mical-1 in complex with Rab10 shows two molecules of Rab10 bound to Mical-1 at different sites. Whereas one Rab protein binds to α-helix 1 and the first half of α-helix 2 (Mical-1$_{918-991}$, binding site 1, blue), the second molecule of Rab10 binds the second half of α-helix 2 and α-helix 3 (Mical-1$_{994-1060}$, binding site 2, green). Upon superimposition of both binding sites, the strong similarity becomes obvious and the helices from both binding sites adopt very similar positions. Furthermore, the interactions are highly similar in both cases as can be seen in the close-up view on the right (similar Rab-interacting residues within binding site 1 and 2 are shown in blue and green, respectively). (b) The strong conservation of interacting residues within both halves of the Mical-1 bMERB domain can also be seen in the sequence alignment of the N- and C-terminal halves. Additionally, the alignment shows that α-helix 1 and the first half of α-helix 2 (binding site 1)

*Figure 6 continued on next page*

*Figure 6 continued*

correspond to the second half of α-helix 2 and α-helix 3 (binding site 2), respectively (the secondary structure is indicated above and below the corresponding sequences, interacting and conserved residues within binding site 1 and 2 are highlighted in blue and green). (c) Whereas the whole bMERB domain of Mical-1 can bind two Rab molecules (left), deletion of either α-helix 1 (middle) or α-helix 3 (right) impairs binding to binding site 1 or 2, respectively. This effect could be shown both by aSEC and ITC (note the enthalpy-driven high-affinity binding site 2 and the entropy-driven lower-affinity binding site 1 that can be clearly distinguished, a schematic drawing of the different constructs is shown within the aSEC graphs). (d) Based on the observations made above, we propose that both binding sites must have evolved from a common ancestor by gene duplication of a 200 bp DNA fragment coding for the original gene product, a single α-hairpin. The fusion lead to the arrangement of the α-helices observed in bMERB domains, with the central α-helix 2 as a continuous connecting helix of both repeats, similar to the architecture of spectrin repeats.

The following figure supplements are available for figure 6:

**Figure supplement 1.** Evolution of the second binding site.

**Figure supplement 2.** Structural comparison of the individual Rab binding sites in Mical-1.

**Figure supplement 3.** Sequence alignment of the bMERB domains examined in this work.

The 3-dimensional structures of the effector domains solved in this work showed a solely α-helical fold common to many other Rab effector proteins (*Oesterlin et al., 2014*). Further comparison with other known Rab:effector structures showed that the main interacting helix in Rab:bMERB complexes (e.g. α-helix 3 in Mical-cL) adopts a similar position to that of the main interacting helix in the structures of Rab27:Slp2-a (*Chavas et al., 2008*), Rab27:Slac2a/melanophilin (*Kukimoto-Niino et al., 2008*) and Rab3:Rabphilin-3a (*Ostermeier and Brunger, 1999*) (see *Figure 5—figure supplement 2* for a comparison). Interestingly, in all three examples, the Arg/Lys contacting Asp45 in our Mical: Rab structures is also conserved in these effector proteins, and the Asp/Asn following this basic residue and contacting Gln61 in Rabs is conserved in both Slp2-a and Slac2a. Intriguingly, these effector domains also display similar binding affinities as bMERB domains towards their cognate Rabs ($K_D =$ 13.4 nM for Rab27:Slp2-a, $K_D =$ 112 nM for Rab27:Slac2-a/melanophilin) (*Fukuda, 2006*), and these are amongst the highest affinities observed for Rab:effector interactions.

Both the biochemical as well as the structural analysis identified a second binding site in Mical-1, Mical-3 and EHBP1L1, thus allowing these effectors to bind Rab proteins in a 1:2 stoichiometry. In contrast to the bMERB domain proteins, Rab:effector complexes that were previously characterized display either a 1:1 or 2:2 stoichiometry, where the 2:2 complexes are usually formed by a central effector dimer with symmetrical binding interfaces on both sites (*Oesterlin et al., 2014*). On the other hand, multivalent Rab effector proteins have been described previously (examples are Rab4 and Rab5 binding to Rabaptin-5 (*Vitale et al., 1998*), Rab6 and Rab11 binding to Rab6IP1 (*Miserey-Lenkei et al., 2007*) or the extreme case of Gcc185 with five sites binding to Rab1a/b, Rab2a/b,

**Table 3.** Non-exhaustive list of conserved interactions between Rab10 and the separate binding sites in Mical-1.

| Mical-1 | | Rab10 |
|---|---|---|
| **Binding site 1** | **Binding site 2** | |
| Glu964 | Glu1030 | Arg70 |
| Lys981 | Arg1044 | Asp45 |
| Asn982 | Asp1045 | Gln61 |
| Leu956 | Leu1011 | Ile42 |
| Val971 | Leu1034 | Ile42 |
| Leu975 | Val1038 | Ile44, Ile74 |
| Val978 | Val1041 | Ile44, Phe46, Trp63 |
| Val985 | Ile1048 | Leu9, Phe46, Ile48 |

Rab6a/b, Rab9a/b, Rab15, Rab27, Rab30, Rab33, Rab35 and Rab36 (*Hayes et al., 2009*). However, all of these effectors contain separate Rab-binding domains, each in turn only binding one Rab protein. The work presented constitutes the first description of two Rab proteins binding a single effector domain.

The separate binding sites within one domain not only represent a novel finding for Rab effector molecules, but also suggest intriguing and hitherto unknown functions of these proteins. Such functions could include linking Rab-decorated vesicles to a target membrane or other vesicles via a central bivalent effector. On the other hand, concerted Rab cascades and feedback loops have been observed with effector domains fused to GEFs or GAPs of one Rab acting upstream or downstream of a second Rab, helping to recruit or remove Rab proteins from a certain membrane (*Pfeffer, 2013*). Bivalent effectors could act in a similar manner in a positive feedback loop, initially being recruited by activated Rab proteins and subsequently helping in the recruitment and stabilization of further Rabs at this site to establish Rab membrane microdomains (*Pfeffer, 2013*). In fact, the presence of one high affinity and one low affinity Rab binding site as observed in some bMERB effectors could further enhance the formation of Rab microdomains: Whereas the Rab bound to the high affinity site would essentially stay bound within physiologically relevant timescales, the additional Rab protein recruited by the low affinity site could dissociate again and recruit another effector molecule via the high affinity site, thus helping to concentrate Rabs within small areas on the membrane.

Additionally and similar to the suggested function of Rab6IP linking Rab6 and Rab11 mediated vesicular trafficking events (*Miserey-Lenkei et al., 2007*), bMERB domain containing effectors might fulfill analogous functions in vesicular trafficking and act as effector Rab hubs. The possible importance of such concerted membrane recruitment cascades of Rabs and other proteins involved in membrane trafficking has been previously highlighted for Mical-L1 connecting Rab35 and Rab8, and this was aptly referred to as a membrane hub (*Rahajeng et al., 2012*; *Giridharan et al., 2012*). Furthermore, recent studies on Mical-L2 dependent GLUT4 translocation showed that trafficking was dependent on a concerted action of Rab8 and Rab13 (*Sun et al., 2010*, *2016*). In another study it was shown that Mical-L1 is recruited to recycling endosomes by Rab35 and subsequently recruits other Rab proteins (Rab8, Rab13 and Rab36) (*Kobayashi et al., 2014*). In this work, the authors concluded that dimerization of Mical-L1 allows a concerted recruitment and binding of two separate Rabs to an effector dimer. However, our data now show how the 2 separate binding sites presumably also present in Mical-L1 and Mical-L2 (see *Figure 6—figure supplement 1*) could help in establishing this concerted action of two Rabs by connecting them via one bivalent effector protein in an (intermediate) 1:1:1 complex, thus explaining for the first time the structural and biochemical basis of the Rab hub function. The strong sequence homology of different bMERB domains including both binding sites and the fact that all Rab8 family proteins reported to interact with Micals/EHBPs are implicated in different steps of endocytic trafficking (*Wandinger-Ness and Zerial, 2014*) as well as previously published data thus points towards an important function of Micals/EHBPs in sorting of endocytic cargo with different destinations in the cell.

Another possible functional implication of the separate binding sites follows from the observation of an auto-inhibited state within Micals. Previous studies have shown that the bMERB domains in Micals can bind to their CH/LIM domains, forming an auto-inhibited intramolecular interaction that can be released by competitive binding of Rab proteins, thus allowing for the interaction with actin only after binding of Rabs (*Sun et al., 2016*; *Sakane et al., 2010*; *Schmidt et al., 2008*). The structural basis as well as the functional significance of this competitive binding will be an interesting topic for future research, especially regarding the binding region at the RBD responsible for auto-inhibition. The two binding sites might therefore also have separate functions, serving as a membrane recruitment site by Rabs via one site (presumably the high-affinity binding site) and release of the auto-inhibition due to competitive binding of the CH/LIM domains and Rabs at the second binding site.

In the last part of the study, we have shown that the two binding sites share a strong similarity and bind Rab proteins via similar residues within the interaction surfaces, thus indicating that the RBD have evolved via duplication of a common ancestor supersecondary structure element (*Figure 6d*) (*Söding and Lupas, 2003*). The underlying single α-hairpin motifs making up the separate Rab-binding sites are known from other Rab-effectors such as Rabenosyn-5 (*Khan and Ménétrey, 2013*; *Eathiraj et al., 2005*), and the resulting fused α-hairpins observed in Micals and EHBPs strongly resemble the architecture of spectrin repeats, each being connected via one continuous

helix (*Han et al., 2007*; *Pascual et al., 1997*). Because of the strong similarity between the repeats of the different isoforms of Micals and EHBPs, but less similarity between both repeats within the bMERB domains (*Figure 6—figure supplement 1b*), we suggest that the duplication must have occurred early in evolution in one common ancestor bMERB domain. The two repeats have since diverged in terms of overall sequence, but the ability to bind Rab proteins has remained in at least some cases, as shown in this study. Exploration of the binding specificity of both binding sites of the bMERB proteins towards Rabs (and possibly other GTPases [*Rahajeng et al., 2012*]) will therefore be of great interest.

## Materials and methods

### Protein expression and purification

Rab proteins were expressed and purified as described previously (Rab1 [*Schoebel et al., 2009*] and other Rab proteins [*Bleimling et al., 2009*]) and preparatively loaded with GppNHp (Guanosine-5'-[β-γ-Imido]-triphosphat) for interaction studies with their effector proteins. For this purpose, the proteins were incubated in the presence of 5 mM EDTA, 5–10% glycerol, three-fold molar excess of GppNHp over the Rab protein and 0.5 units alkaline phosphatase per mg Rab protein for 2 hr at 20°C or at 4°C overnight. Subsequently the proteins were purified via gel filtration in a buffer containing 20 mM Hepes pH 7.5, 50 mM NaCl, 2 mM DTE, 2 mM $MgCl_2$ and 10 μM GppNHp. bMERB domains (amino acid boundaries and Uniprot accession IDs are shown in *Figure 6—figure supplement 3*) were cloned into a modified pET19 expression vector and proteins were expressed in *E. coli* BL21 DE3 RIL cells (growth at 37°C to $OD_{600\ nm}$ = 0.8–1.0, stored at 4°C for 30 min, expression was induced by addition of 0.3–0.5 mM IPTG and cells were grown for 14–18 hr at 20°C). Subsequently the proteins were purified by $Ni^{2+}$-affinity chromatography, cleavage of the $His_6$-tag with TEV-protease and a second $Ni^{2+}$-affinity purification. Final purification was achieved by gel filtration (Rabs: 20 mM Hepes pH 7.5, 50 mM NaCl, 2 mM DTE, 2 mM $MgCl_2$, 10 μM GDP or GppNHp; Micals: 20 mM Hepes pH 7.5, 50 or 100 mM NaCl, 2 mM DTE). In order to express the selenomethionine labeled version of the coiled coil domain of Mical-3, the methionine biosynthesis inhibition method (*Van Duyne et al., 1993*) was used. The labeled protein was purified as described above. FTase and GGTase I were purified as described previously (*Dursina et al., 2006*; *Kalinin et al., 2001*).

For prenylation, $EHBP1_{1047-1231}$/$EHBP1L1_{1340-1523}$, substrate (farnesyl pyrophosphate (FPP) or geranylgeranyl pyrophosphate (GGPP); Sigma) and prenyltransferase (FTase or GGTase I) were mixed in a 1:5:0.5 ratio in buffer containing 25 mM Hepes pH 7.2, 40 mM NaCl and 2 mM $MgCl_2$ and incubated for 3 hr at room temperature. To check the extent of prenylation, samples were analyzed by ESI-MS.

### Analytical size exclusion chromatography

Complex formation of Rab proteins preparatively loaded with GppNHp and the bMERB effector domains was assessed by analytical size exclusion chromatography (aSEC). The effector domains were used at a concentration of 113 μM, Rab proteins were used at concentrations of 130 μM (Rab: effector stoichiometry of 1.2:1), 180 μM (1.6:1) or 250 μM (2.2:1) and 30 μl of the protein solutions were injected into a Superdex S75 10/300 GL gel filtration column (flow rate 0.5 ml/min, detection of absorption at 280 nm, buffer: 20 mM Hepes pH 7.5, 50 mM NaCl, 2 mM DTE, 2 mM $MgCl_2$).

### Isothermal titration calorimetry

Protein-protein interaction was studied by ITC using an $iTC_{200}$ microcalorimeter (MicroCal). Measurements were performed in buffer containing 20 mM Hepes pH 7.5, 50 mM NaCl, 2 mM $MgCl_2$ and 1 mM tris (2-carboxymethyl) phosphine (TCEP) at 25°C and for every experiment a technical replicate was performed. 400 μM Rab was titrated into the cell containing 20–40 μM of Mical. Data were analyzed with Origin (Version 7.0, MicroCal).

### X-ray crystallography

Initial crystallization conditions for single effector proteins and all protein complexes described here were determined with the JSG Core I-IV, Pact and Protein Complex suites from Qiagen. The sitting-

drop vapor diffusion method was used, with a reservoir volume of 70 µl and a drop volume of 0.1 µl protein (15–25 mg ml$^{-1}$) and 0.1 µl reservoir solution at 20°C. The best conditions were then optimized using the hanging-drop vapor diffusion method in order to obtain well diffracting crystals. The seleno-L-methionine labelled Mical-3$_{1841–1990}$ was finally crystallized in 0.1 M Tris pH 7.0 und 45 – 50% PEG 200 (protein concentration 5.5 – 11 mg/ml). Mical-cL$_{534–683}$ in a 1 to 1 complex ratio with all tested Rab proteins crystallized in similar conditions. The complex with Rab1b$_{Fl}$ was crystallized in 0.1 M bis-tris-propane pH 8.4–8.6, 0.2 M tri-sodium citrate and 20–22% (w/v) PEG 3350, with Rab8a$_{Fl}$ in 0.1 M bis-tris-propane pH 8.3–8.7, 0.2 M tri-sodium citrate and 18–20% (w/v) PEG 3350 and finally with Rab10$_{Fl}$ in 0.2–0.3 M sodium acetate and 18–22% (w/v) PEG 3,350. The hybrid Rab1b$_{R8N}$ (chimera) in complex with Mical-cL$_{534–683}$ was crystallized in 0.1 M bis-tris pH 7.5, 0.2 M sodium malonate and 20% (w/v) PEG3350. Mical-1$_{918–1067}$ crystallized with Rab10$_{1-175}$ in a 1 to 2 ratio under the following conditions: 0.1 M imidazole pH 7.6–8.0 and 6–10% (w/v) PEG 8,000.

Best diffracting crystals were flash-cooled in liquid nitrogen and diffraction data were collected on beamline X10SA at the Swiss Light Source (Paul Scherrer Institute, Villigen, Switzerland) and processed with XDS (*Kabsch, 2010*). The structure of Mical-3$_{1841–1992}$ was solved by the single anomalous diffraction method using data collected at the selenium absorption edge. Initial phases and an initial model were obtained with PHENIX AutoSol (*Adams et al., 2010*). All protein complex structures were solved by the maximum likelihood molecular replacement method using the structures of Mical-3$_{1841–1992}$, Mical-cL$_{534–683}$, Rab1b (pdb id 3nkv) and Rab8a (pdb id 4lhw) as search models. The initial structure models were completed by hand in Coot (*Emsley et al., 2010*) and refined with phenix.refine (*Adams et al., 2010*) or Refmac5 (*Murshudov et al., 1997*) of the CCP4 package (*Winn et al., 2011*) using the TLS option. Data collection and refinement statistics, as well the Protein Data Bank accession numbers of each presented structure are summarized in *Table 1*.

## Fluorescence microscopy

Rab constructs were cloned into pmCherry vector using the XhoI and BamHI restriction sites and to obtain active Rab mutants (Rab8$_{Q67L}$, Rab10$_{Q68L}$) quick change mutagenesis was performed. Further EHBP1$_{1047-1231}$, EHBP1$_{1047-1227}$ (missing the CaaX-box), EHBP1L1$_{1340-1523}$ and EHBP1L1$_{1340–1519}$ (missing the CaaX-box) constructs were cloned into pEGFP(C1) vector between EcoRI and SalI sites.

Cos7 cells were maintained in DMEM medium supplemented with 10% fetal bovine serum, 2 mM L-glutamine and penicillin/streptomycin at 37°C in the presence of 5% CO$_2$. Cells were grown on a coverslip in 6 well plate until they reached 60–70% confluency and transiently transfected using polyethylenimine (PEI, Polysciences,Inc, 3:1 PEI:DNA (12:4 µg). Expression was checked 20–24 hr post transfection. Cells were fixed with 3.7% paraformaldehyde in PBS for 15 min at room temperature. Unreacted paraformaldehyde was quenched with 100 mM glycine in PBS for 15 min. After washing with PBS, cover slips were mounted on glass slides with SlowFade Gold antifade reagent (Invitrogen). Images were taken using a Leica TCS SP8 confocal microscope, 63 × 1.4 NA HC PL APO CS2 oil immersion objective. For Delta CaaX constructs single plane images were taken and for 3D reconstitutions, three dimensional stacks with 0.3-µm steps were acquired. Images from the all focal planes were rendered as a single maximum-intensity projection using Leica software.

## Acknowledgements

We would like to thank Prof. Andrei Lupas (MPI Tübingen) for helpful discussions regarding the evolution of the bMERB domains. Prof. Christian Herrmann (Ruhr-University Bochum) is acknowledged for the generous possibility to measure ITC data using the Auto-iTC200 in his group. Nathalie Bleimling is acknowledged for invaluable technical assistance. We thank the Swiss Light Source (SLS) X10SA beamline staff for the possibility to collect data at the Paul Scherrer Institute, Villigen, Switzerland. Many thanks go to Georg Holtermann for his technical support in x-ray crystallography and inhouse data collection. Some expression plasmids used in this work were cloned by the Dortmund Protein Facility (DPF). AI acknowledges support by the Deutsche Forschungsgemeinschaft (SFB1035, project B05) and the authors thank the Max-Planck-Society and the Deutsche Forschungsgemeinschaft (SFB642, project A4) for financial support.

## Additional information

### Funding

| Funder | Grant reference number | Author |
| --- | --- | --- |
| Max-Planck-Gesellschaft | | Amrita Rai<br>Roger S Goody<br>Emerich Mihai Gazdag<br>Matthias P Müller |
| Deutsche Forschungsgemeinschaft | SFB1035, project B05 | Aymelt Itzen |
| Deutsche Forschungsgemeinschaft | SFB642, project A4 | Roger S Goody<br>Matthias P Müller |

The funders had no role in study design, data collection and interpretation, or the decision to submit the work for publication.

### Author contributions

AR, Performed the x-ray crystallographic experiments, Performed the in vivo localization experiments, Performed the biochemical and biophysical experiments, Conception and design, Analysis and interpretation of data, Drafting or revising the article; AO, JC, YF, TF, Performed the biochemical and biophysical experiments; AI, Conception and design, Drafting or revising the article; RSG, Conception and design, Analysis and interpretation of data, Drafting or revising the article; EMG, MPM, Performed the biochemical and biophysical experiments, Performed the x-ray crystallographic experiments, Conception and design, Analysis and interpretation of data, Drafting or revising the article

### Author ORCIDs

Roger S Goody, http://orcid.org/0000-0002-0772-0444
Matthias P Müller, http://orcid.org/0000-0002-1529-8933

## Additional files

### Major datasets

The following datasets were generated:

| Author(s) | Year | Dataset title | Dataset URL | Database, license, and accessibility information |
| --- | --- | --- | --- | --- |
| Rai A, Oprisko A, Campos J, Fu Y, Friese T, Itzen A, Goody RS, Gazdag EM, Mueller MP | 2016 | Structure of the bMERB domain of Mical-3 | http://www.rcsb.org/pdb/explore/explore.do?structureId=5SZG | Publicly available at the RCSB Protein Data Bank (accession no. 5SZG) |
| Rai A, Oprisko A, Campos J, Fu Y, Friese T, Itzen A, Goody RS, Gazdag EM, Mueller MP | 2016 | Structure of human N-terminally engineered Rab1b in complex with the bMERB domain of Mical-cL | http://www.rcsb.org/pdb/explore/explore.do?structureId=5SZK | Publicly available at the RCSB Protein Data Bank (accession no. 5SZK) |
| Rai A, Oprisko A, Campos J, Fu Y, Friese T, Itzen A, Goody RS, Mueller MP, Gazdag EM | 2016 | Structure of human Rab10 in complex with the bMERB domain of Mical-1 | http://www.rcsb.org/pdb/explore/explore.do?structureId=5LPN | Publicly available at the RCSB Protein Data Bank (accession no. 5LPN) |
| Rai A, Oprisko A, Campos J, Fu Y, Friese T, Itzen A, Goody RS, Mueller MP, Gazdag EM | 2016 | Structure of human Rab10 in complex with the bMERB domain of Mical-cL | http://www.rcsb.org/pdb/explore/explore.do?structureId=5SZJ | Publicly available at the RCSB Protein Data Bank (accession no. 5SZJ) |

| Rai A, Oprisko A, Campos J, Fu Y, Friese T, Goody RS, Mueller MP, Gazdag EM | 2016 | Structure of human Rab1b in complex with the bMERB domain of Mical-cL | http://www.rcsb.org/pdb/explore/explore.do?structureId=5SZH | Publicly available at the RCSB Protein Data Bank (accession no. 5SZH) |
| Rai A, Oprisko A, Campos J, Fu Y, Friese T, Itzen A, Goody RS, Mueller MP, Gazdag EM | 2016 | Structure of human Rab8a in complex with the bMERB domain of Mical-cL | http://www.rcsb.org/pdb/explore/explore.do?structureId=5SZI | Publicly available at the RCSB Protein Data Bank (accession no. 5SZI) |

The following previously published datasets were used:

| Author(s) | Year | Dataset title | Dataset URL | Database, license, and accessibility information |
|---|---|---|---|---|
| Mueller MP, Peters H, Blankenfeldt W, Goody RS, Itzen A | 2010 | Crystal structure of Rab1b covalently modified with AMP at Y77 | http://www.rcsb.org/pdb/explore/explore.do?structureId=3nkv | Publicly available at the RCSB Protein Data Bank (accession no. 3NKV) |
| Guo Z, Hou XM, Goody RS, Itzen A | 2013 | Crystal structure of Rab8 in its active GppNHp-bound form | http://www.rcsb.org/pdb/explore/explore.do?structureId=4lhw | Publicly available at the RCSB Protein Data Bank (accession no. 4LHW) |

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
