## [Decision Letter]

Thank you for submitting your article "bMERB domains are bivalent Rab8 effectors evolved by gene duplication" for consideration by *eLife*. Your article has been reviewed by three peer reviewers, one of whom, Reinhard Jahn, is a member of our Board of Reviewing Editors and the evaluation has been overseen by John Kuriyan as the Senior Editor.

The reviewers have discussed the reviews with one another and the Reviewing Editor has drafted this decision to help you prepare a revised submission.

This manuscript includes a thorough biochemical and structural characterization of the interaction between Rab8 family proteins and an effector domain shared by the MICAL and EHBP1 protein families. All reviewers agree that the work is of very high quality and that the manuscript is excellent. For these reasons, the reviewers do not request additional experiments but rather ask for considering the (rather minor) issues raised by Reviewers 2 and 3. This also includes a possible change of the title of the manuscript as suggested by Reviewer 3. Please note, however, that none of these suggested changes is mandatory for acceptance.

Reviewer #2:

a) The authors state in the Abstract that their study is "the first to thoroughly characterise a Rab effector domain that contains two separate binding sites". This is potentially confusing as of course several Rab effectors are dimeric and so contain two separate binding sites in the dimer. The text needs to be clarified.

b) Figure 1. Are all eight proteins from separate genes or are some alternatively spliced transcripts from the same gene? This should be made clear in the text and figure.

c) Figure 3. The Mical-3 bMERB domain is a dimer in the crystal. Is it also a dimer in solution? Also, is it still a dimer when co-crystalised with the Rabs in Figure 5 and Figure 6? This is an important point which needs to be discussed clearly in the text.

d) Results section: The "non-exhaustive list of conserved interactions" could instead be presented as a table rather than in a long sentence.

*Reviewer #3:*

As currently written the title implies that bMERB8 proteins are Rab8 effectors, rather than effectors for the wider Rab8 family of GTPases (8/10/13/15). The data presented in Table 1, although not complete, indicate that Rab13 and Rab15 might be the preferential partners for Mical-cL. Figure 6 shows a structure with Rab10 and Mical-1 rather than Rab8. In the absence of further functional data linking the various Rabs to the effectors, the title should be made more general and refer to the Rab8 subfamily of GTPases. The Abstract should also specifically mention which Rab GTPases the authors have looked at: Rab8/10/13/15. Writing "Rab8 family proteins" might be interpreted as meaning only Rab8A and 8B by less careful readers.

I don't follow the logic of the statement at the end of the Abstract: "a Rab effector domain that contains two separate binding sites, allowing Micals and EHBPs to bind two Rabs simultaneously and thus suggesting novel and previously unknown functions of these effector molecules in endosomal trafficking". If these were GTPases of different families, e.g. Rab and Rho or Ras, then the argument would be clear. However, binding multiple Rabs still implies a link to functions in membrane trafficking rather than other cellular processes. What is novel or unknown in that sense? I agree with the view that I suspect informs this statement, which is that we don't really understand what the Mical/EHBP proteins do, or how Rab-binding controls their function. Multivalent Rab binding might allow one effector integrate inputs from different Rabs at one membrane or perhaps different membranes bearing different Rabs. Perhaps something along these lines could be referred to at the end of the Abstract.

Why was Rab35 not investigated? Given the extensive characterisation of interactions with the related Rabs this is an obvious omission.

EHBP1 prenylation data and membrane targeting in Figure 3 are interesting but don't fit with the main thrust of the work that is focussed on the Rab-effector interaction. This would be better as supplemental information for Figure 1.

Figure 5—figure supplement 1 and Figure 6—figure supplement 2. Labelling of residues in the bMERB8 domains are too small to read.

---

## [Author Response]

*This manuscript includes a thorough biochemical and structural characterization of the interaction between Rab8 family proteins and an effector domain shared by the MICAL and EHBP1 protein families. All reviewers agree that the work is of very high quality and that the manuscript is excellent. For these reasons, the reviewers do not request additional experiments but rather ask for considering the (rather minor) issues raised by Reviewers 2 and 3. This also includes a possible change of the title of the manuscript as suggested by Reviewer 3. Please note, however, that none of these suggested changes is mandatory for acceptance.*

*Reviewer #2:*

*a) The authors state in the Abstract that their study is "the first to thoroughly characterise a Rab effector domain that contains two separate binding sites". This is potentially confusing as of course several Rab effectors are dimeric and so contain two separate binding sites in the dimer. The text needs to be clarified.*

In order to clarify this point we have rephrased the sentence in the Abstract:

“This study is the first to thoroughly characterise a Rab effector protein that contains two separate Rab binding sites within a single domain, allowing […]".

*b) Figure 1. Are all eight proteins from separate genes or are some alternatively spliced transcripts from the same gene? This should be made clear in the text and figure.*

They are all from separate genes. To clarify this we have added the following information in the figure legend:

“For proteins with multiple known splice variants, domain boundaries are indicated for isoform 1 (Mical-1: Uniprot ID Q8TDZ2, genomic location 6q21; Mical-L1: Uniprot ID Q8N3F8, genomic location 22q13.1; Mical-L2: Uniprot ID Q8IY33, genomic location 7p22.3; Mical-3: Uniprot ID Q7RTP6, genomic location 22q11.21; Mical-cL: Uniprot ID Q6ZW33, genomic location 11p15.3; EHBP1: Uniprot ID Q8NDI1, genomic location 2p15; EHBP1L1 Uniprot ID Q8N3D4, genomic location 11q13.1).”

*c) Figure 3. The Mical-3 bMERB domain is a dimer in the crystal. Is it also a dimer in solution? Also, is it still a dimer when co-crystalised with the Rabs in Figure 5 and Figure 6? This is an important point which needs to be discussed clearly in the text.*

Indeed, both Mical-3 and Mical-cL appear to form dimers in solution, whereas Mical-1 does not. However, we can currently not comment on any physiological significance of this. We have added one paragraph in order to clarify this point:

“Interestingly, the Rab-binding interface in Mical-cL has a substantial overlap with the dimer interface observed in the structure of Mical-3 above. Additionally, even though all Mical constructs used have a similar molecular weight of ~18 kDa, whereas Mical-1 runs as an apparent monomer in aSEC and binding of a Rab protein induces a clear shift to higher molecular weight, both Mical-3 and Mical-cL run as apparent dimers in aSEC and binding of a Rab protein disrupts the dimer, thus not leading to a shift in retention time upon complex formation (Figure 2—figure supplement 1). It is however not clear at this point whether the dimer formation of Mical-3 and Mical-cL and the disruption of the dimer upon Rab-binding is of functional significance.”

d) Results section: The "non-exhaustive list of conserved interactions" could instead be presented as a table rather than in a long sentence.

According to the reviewers’ suggestion we have added Table 3 and changed the manuscript accordingly:

“A non-exhaustive list and a close-up view of several of these interactions is shown in Table 3 and Figure 6—figure supplement 2, respectively.”

*Reviewer #3:*

*As currently written the title implies that bMERB8 proteins are Rab8 effectors, rather than effectors for the wider Rab8 family of GTPases (8/10/13/15). The data presented in Table 1, although not complete, indicate that Rab13 and Rab15 might be the preferential partners for Mical-cL. Figure 6 shows a structure with Rab10 and Mical-1 rather than Rab8. In the absence of further functional data linking the various Rabs to the effectors, the title should be made more general and refer to the Rab8 subfamily of GTPases. The Abstract should also specifically mention which Rab GTPases the authors have looked at: Rab8/10/13/15. Writing "Rab8 family proteins" might be interpreted as meaning only Rab8A and 8B by less careful readers.*

According to the reviewers’ suggestion we have updated the title (“bMERB domains are bivalent Rab8 family effectors evolved by gene duplication”) and additionally modified the Abstract:

“Within our study, we show that these effectors display a strong preference for Rab8 family proteins (Rab8, 10, 13 and 15) and that some of the effector domains can bind two Rab proteins […]”.

*I don't follow the logic of the statement at the end of the Abstract: "a Rab effector domain that contains two separate binding sites, allowing Micals and EHBPs to bind two Rabs simultaneously and thus suggesting novel and previously unknown functions of these effector molecules in endosomal trafficking". If these were GTPases of different families, e.g. Rab and Rho or Ras, then the argument would be clear. However, binding multiple Rabs still implies a link to functions in membrane trafficking rather than other cellular processes. What is novel or unknown in that sense? I agree with the view that I suspect informs this statement, which is that we don't really understand what the Mical/EHBP proteins do, or how Rab-binding controls their function. Multivalent Rab binding might allow one effector integrate inputs from different Rabs at one membrane or perhaps different membranes bearing different Rabs. Perhaps something along these lines could be referred to at the end of the Abstract.*

Indeed we agree that they presumably have a function in endosomal vesicular trafficking (we stated “suggesting novel and previously unknown functions of these effector molecules in endosomal trafficking."). However, the presence of two separate binding sites implies that (for as yet unknown reasons) these effectors bind two different Rab proteins simultaneously. As we discuss later in the manuscript, this might include functions “in sorting cargo and/or linking different endosomal trafficking pathways regulated by different Rab proteins”, however we have no experimental data at this point that would allow us to do more than speculate.

*Why was Rab35 not investigated? Given the extensive characterisation of interactions with the related Rabs this is an obvious omission.*

Rab35 was actually investigated, however upon preparative nucleotide exchange of Rab35 with GppNHp, the protein selectively bound and enriched an impurity present in the commercially available GppNHp, the hydrolysis product GppNH_2_. Because of these problems with preparative loading of the protein with GppNHp the data was not included in the first version of the paper. We have however now included data for Rab35 and Mical-cL in Figure 2—figure supplement 2, including the data about preparative GppNHp binding.

We also updated the main manuscript (including Table 1) accordingly:

“Using Mical-cL as one representative of the bMERB family, we saw that all members of the Rab8 family bound Mical-cL with high nanomolar affinities, compared to Rab1 (K_D_ = 5.2 µM) and Rab35 (K_D_ = 1.8 µM; Table 1 and Figure 2—figure supplement 2)”.

“Mical-cL (Figure 5). Additionally, Rab35 contains an Arg residue within this N-terminal region and also displays a slightly higher affinity towards Mical-cL compared to Rab1 (Figure 2—figure supplement 2). We therefore tested whether these N-terminal […]”.

*EHBP1 prenylation data and membrane targeting in Figure 3 are interesting but don't fit with the main thrust of the work that is focussed on the Rab-effector interaction. This would be better as supplemental information for Figure 1.*

We feel that the information does fit the main thrust of the work since we argue that DUF3585 bMERB domains are Rab8 family effectors. Therefore, *in vivo* localization studies showing that EHBP1 and EHBP1L1 colocalize with Rab8 family members at endosomes strengthens this point and we would therefore like to keep Figure 3 as it is.

*Figure 5—figure supplement 1 and Figure 6—figure supplement 2. Labelling of residues in the bMERB8 domains are too small to read.*

We have increased the font size in the corresponding figures.

In addition to the changes mentioned above we have added the following minor changes:

The citation for the XDS program package was updated;Amino acid boundaries for Mical-1 and the wavelength of data collection were included in the data statistics table for x-ray crystallography;Scale bars have been added in Figure 3;The acknowledgments have been updated;Minor changes in the Abstract were necessary to comply with the word limit.